# Comparative Study of SARS-CoV-2, SARS-CoV-1, MERS-CoV, HCoV-229E and Influenza Host Gene Expression in Asthma: Importance of Sex, Disease Severity, and Epithelial Heterogeneity

**DOI:** 10.3390/v13061081

**Published:** 2021-06-05

**Authors:** Mackenzie E. Coden, Lucas F. Loffredo, Hiam Abdala-Valencia, Sergejs Berdnikovs

**Affiliations:** 1Division of Allergy and Immunology, Department of Medicine, Northwestern University Feinberg School of Medicine, Chicago, IL 60611, USA; mackenzie.coden@gmail.com (M.E.C.); lfl2128@cumc.columbia.edu (L.F.L.); 2Division of Pulmonary and Critical Care, Department of Medicine, Northwestern University Feinberg School of Medicine, Chicago, IL 60611, USA; h-abdala-valencia@northwestern.edu

**Keywords:** COVID-19, SARS, HCoV-229E, MERS, influenza, virus, epithelium, asthma, allergy, inflammation, sexual dimorphism, gene expression

## Abstract

Epithelial characteristics underlying the differential susceptibility of chronic asthma to SARS-CoV-2 (COVID-19) and other viral infections are currently unclear. By revisiting transcriptomic data from patients with Th2 low versus Th2 high asthma, as well as mild, moderate, and severe asthmatics, we characterized the changes in expression of human coronavirus and influenza viral entry genes relative to sex, airway location, and disease endotype. We found sexual dimorphism in the expression of SARS-CoV-2-related genes ACE2, TMPRSS2, TMPRSS4, and SLC6A19. ACE2 receptor downregulation occurred specifically in females in Th2 high asthma, while proteases broadly assisting coronavirus and influenza viral entry, TMPRSS2, and TMPRSS4, were highly upregulated in both sexes. Overall, changes in SARS-CoV-2-related gene expression were specific to the Th2 high molecular endotype of asthma and different by asthma severity and airway location. The downregulation of ACE2 (COVID-19, SARS) and ANPEP (HCoV-229E) viral receptors wascorrelated with loss of club and ciliated cells in Th2 high asthma. Meanwhile, the increase in DPP4 (MERS-CoV), ST3GAL4, and ST6GAL1 (influenza) was associated with increased goblet and basal activated cells. Overall, this study elucidates sex, airway location, disease endotype, and changes in epithelial heterogeneity as potential factors underlying asthmatic susceptibility, or lack thereof, to SARS-CoV-2.

## 1. Introduction

As the COVID-19 pandemic continues to unfold worldwide, new insights are coming to light about the mechanisms and epidemiology of this disease. One of the most puzzling recent observations is an apparent lack of association of SARS-CoV-2 infection with asthma. Reports from Wuhan, China revealed that asthma was not a risk factor for severity and mortality in adult COVID-19 patients—the prevalence of asthma in patients with COVID-19 was only 0.9%, which is substantially lower than that in the adult population of Wuhan (6.4%) [1]. In another study of 140 patients infected with SARS-CoV-2 in Wuhan, none presented with asthma or other allergic diseases [2]. A study by Mahdavinia et al. [3] in a large cohort of adult COVID-19 cases in the United States showed that asthma was significantly associated with longer intubation but was not associated with a higher rate of hospitalization, death, or acute respiratory distress syndrome among COVID-19 patients. This is surprising because respiratory viral infections generally associate with asthma exacerbations and more serious illness. Recent studies provide an explanation for this by showing that asthma and controlled allergen exposures are associated with significant reductions in epithelial expression of ACE2, the entry receptor for SARS-CoV-2 [4,5]. Specifically, ACE2 expression was lowest in individuals with both high levels of allergic sensitization and asthma [4]. Another study confirmed this observation by showing that Type 2 cytokine IL-13 decreases expression of ACE2 in bronchial epithelium, thereby providing one mechanistic link to Th2 inflammation [6]. However, it is not known whether susceptibility to different viruses correlates with the expression of viral receptors in asthmatic airway epithelial cells. Moreover, asthma is a highly heterogeneous disease with potentially multiple factors underlying susceptibility to infection. In this comparative study, we bring novelty to this subject by comparing host entry gene expression of multiple viral species and emphasizing the importance of acknowledging sex differences, disease severity, and epithelial heterogeneity in studies of asthmatic susceptibility to viral infection.

## 2. Materials and Methods

### 2.1. Transcriptomic Data

In this study, we used the following two independent gene expression datasets downloaded from Gene Expression Omnibus (GEO, NCBI): (1) GSE67472, providing data for gene expression in bronchial epithelial brushings from healthy controls and subjects with Th2 low and Th2 high asthma and (2) GSE41861, providing epithelial brushing data for mild, moderate, and severe asthma in both upper and lower airways. The study comparing Th2 low and Th2 high asthma by Woodruff et al. [7] is the most highly cited study providing a basis for molecular endotyping of asthma patients based on patterns of Type 2 inflammation. Clinical descriptions of asthma patients and airway sampling procedures can be found in GEO Datasets and the original publication of this study [7]. Gene annotations were updated using technology-corresponding annotation libraries in R. Each GEO dataset underwent consistent handling, implementing standard contrast fits on series matrix file data uploaded to GEO by the original study groups. This preserved the original normalization protocols to maintain consistency with gene signatures published by original studies. 

### 2.2. Clustering of Epithelial and Viral Entry Genes

Single cell analysis data used to generate an epithelial subset gene list for unsupervised clustering of epithelial and viral entry genes were from a study by Vieira Braga et al. [7], providing a cellular census of the human lung transcriptome in health and asthma. Exact genes and epithelial subset clustering structure in health and asthma can be found in the original Vieira Braga publication. Viral entry genes were selected based on published works on human coronavirus and influenza entry genes in epithelial cells. An epithelial-viral entry gene expression data set was then assembled from GSE67472 data and included both males and females; and healthy, Th2 low, and Th2 high individuals. The resulting data set was subjected to hierarchical clustering using the one minus Pearson correlation metric and average linkage method. A similarity matrix was derived from clustered data using the Pearson correlation metric. All procedures were performed in MORPHEUS, https://software.broadinstitute.org/morpheus (accessed on September 1 2020, Broad Institute), matrix visualization and analysis software. The clustering structure in this publication is nearly an exact match to the study mentioned above, demonstrating alignment in methodology and biological message regarding epithelial heterogeneity.

### 2.3. Statistical Analysis and Data Visualization

All graphs and statistical analyses were carried out in GraphPad Prism (GraphPad) and Systat 13 (Systat Software, Inc., Chicago, IL, USA). Principal Component Analysis (PCA) was executed in PAST 2.17 software using covariance matrices and singular value decomposition. Statistical significance was determined by ANOVA followed by Tukey’s post-hoc pairwise testing. All data are represented as mean ± S.E.M. An alpha level of 0.05 was used as a significance cut-off. Marginal values (0.05 > *p* < 0.1) are also shown.

### 2.4. Data Availability

All raw data used in this study, including series matrix file data, can be accessed in GEO under the accession numbers GSE67472 and GSE41861. scRNA-seq data that support the findings of this study can be found in the study mentioned above by Vieira Braga et al. [8].

## 3. Results and Discussion

We revisited publicly available transcriptomic data from two independent studies of bronchial and nasal epithelium for an unbiased assessment of epithelial expression of SARS-CoV-2, influenza, and other human coronavirus related genes in (1) Th2 low versus Th2 high asthma [7] in bronchial epithelial brushings (GSE67472) and (2) expression of these genes in mild, moderate, and severe asthma in both upper and lower airways (GSE41861). We found that dysregulation of these genes in asthma is both sexually dimorphic and airway location-dependent. For SARS-CoV-2 related genes, we found marginally significant downregulation of viral entry receptor ACE2 in females in Th2 high asthma and significant downregulation in males in mild asthma in the upper airway (6.9% decrease) (Figure 1a). Expression of specific ACE2-partnering membrane protein SLC6A19 (B^0^AT1) was also found to be downregulated in males in mild asthma (lower airway) and moderate asthma (upper airway) (Figure 1a). Interestingly, we found an opposite expression pattern for two related membrane-bound serine proteases, TMPRSS2 and TMPRSS4, which promote SARS-CoV-2 virus epithelial entry by processing the spike glycoprotein to release viral contents into the host cell cytosol. Expression of both proteases was significantly upregulated, specifically in Th2 high asthma (Figure 1a). We also found upregulation of TMPRSS4 in females in severe asthma (lower and upper airway) and TMPRSS2 in males in mild and moderate asthma (lower airway) (Figure 1a). Expression of another protease suspected in assisting viral entry, FURIN, was not found to be significantly different in any dataset (data not shown). Such differences due to sex deserve further investigation, as COVID-19 resulted in higher mortality and severity in males [9]. Downregulation of ACE2 expression predominantly in mild asthma is also interesting, as asthma is a very heterogeneous disease. The Center for Disease Control (CDC) currently lists moderate to severe asthma as a potential risk factor for severe COVID-19 illness (last updated on 7 April 2021). Whether the downregulation of ACE2 confers a protective effect in mild asthma instead of more severe forms of the disease remains to be investigated. In support, a study by Kermani et al. [10] found an increased rather than decreased expression of ACE2 in sputum of subjects with severe neutrophilic asthma compared to mild-moderate asthma, which was positively associated with oral corticosteroid use and male gender. Expression of ACE2 was also significantly upregulated in the airway in patients with chronic obstructive pulmonary disease (COPD) compared to healthy subjects [11]. A study by Kasela et al. [12] further suggested that heterogeneity of asthma and associated host factors (genetics, systemic disease, steroid use) likely determine the expression of SARS-CoV-2-related genes. Of note, the use of inhaled corticosteroids in asthma was associated with lower expression of ACE2 and TMPRSS2 [13]. However, severe asthma patients are most likely to use corticosteroids, yet we found significant ACE2 downregulation only in mild asthma. We performed a Principal Component Analysis (PCA) analysis of covariance in several virus entry genes (ANPEP, DPP4, CTSL, CTSB, ST3GAL4, ST6GAL1) and associated viral immunity genes (IRAK3, IDO1, NOS2, OAS1, MX1, TNFSF10). The PCA demonstrated that dysregulation of viral-associated immunity, in general, is characteristic of the Th2 high but not Th2 low subset of asthma patients (Figure 1b).

We further performed a comparative analysis of SARS-CoV-2 genes with genes promoting the entry of other human coronaviruses and influenza. We found that ANPEP, the entry receptor for HCoV-229E, was significantly downregulated specifically in Th2 high asthma, as well as downregulated in females in mild, moderate, and severe asthma (lower airway) (Figure 2a). Interestingly, HCoV-229E appears to be the human coronavirus least associated with asthma exacerbations. Studies have failed to find HCoV-229E in either adult or childhood episodes of asthma exacerbation [14]. However, asthma was a risk for MERS-CoV (Middle Eastern Respiratory Syndrome) infection. In one study, 31% of those with MERS had asthma [15]. DPP4, the entry receptor for MERS-CoV, was downregulated in females in Th2 low asthma (trending up in Th2 high asthma) and significantly upregulated in females in mild and moderate asthma (lower airway) (Figure 2a). Two genes that promote the synthesis of receptors for most human influenza strains, ST3GAL4 and ST6GAL1, were specifically upregulated in Th2 high inflammation in both sexes (Figure 2a). SARS-Cov-1 (SARS) also relies on the ACE2 receptor as an entry mechanism. The SARS (SARS-CoV-1) outbreak also did not affect asthma exacerbations in children [16].

Following the SARS outbreak in 2002–2003, multiple studies examined mechanisms of SARS infectivity in epithelial cells. Multiple studies confirmed that ACE2 receptor expression positively correlated with the differentiation state of bronchial epithelial cells. Importantly, undifferentiated human airway epithelial cells expressing low levels of ACE2 receptor were poorly infected with SARS-CoV-1, while differentiated cells with higher expression of ACE2 were more readily infected [17]. Given that asthma and other allergic diseases associate with loss of epithelial differentiation [18], this prompted us to examine the possibility that Th2 inflammation modulates viral susceptibility by affecting the differentiation of epithelial cells. Few studies have looked at SARS-CoV-2 gene expression in airway epithelial cells by single cell RNA-seq, mostly at a healthy baseline. At the protein and gene expression level, ACE2 primarily localizes to club (Clara) cells in bronchial epithelium and ATII in alveoli in the mouse lung [19]. The Human Cell Atlas Consortium [20] summarized ACE2 expression data across multiple single cell studies of human upper and lower airways. It showed that in the nasal epithelium, ACE2 expression was the highest in ciliated and goblet cells. In the lower airway, the highest expression was found in club cells; and in the lung parenchyma (alveoli), it was primarily found in ATII cells. Another human study reports the expression of ACE2 in the lung in the club and ciliated cells [21]. In a single cell study in the human airway, ACE2 was also detected primarily in ATII cells and club cells [22]. Together, this shows that viral entry gene expression is restricted to specific epithelial cell subsets.

Unfortunately, no studies are comparing viral entry genes regarding epithelial heterogeneity across Th2 low and Th2 high asthma phenotypes. By comparing known markers of epithelial cell subsets between healthy, Th2 low, and Th2 high asthma (control expression of Th2 cytokines is shown in Appendix A), we found evidence for a remarkable reshaping of epithelial heterogeneity in Th2 high asthma (Figure 2b). We found an increase in markers of basal cycling, basal activated, goblet, mucous ciliated cells, and ionocytes in allergic inflammation. At the same time, there was a significant decrease in markers of ciliated cells, submucosal glands, club (Clara), and tuft cells. This pattern of epithelial dysregulation was highly specific to Th2 high inflammation, confirming at high resolution a loss of epithelial differentiation and specialized cell subsets expressing the SARS-CoV-2 receptor (club cells, ciliated cells). We sought to gain further insights into whether the expression of viral-related genes may correspond to epithelial heterogeneity patterns. We clustered top markers describing each epithelial subset (guided by a recently published single cell study characterizing novel airway epithelial cell subsets in asthma [8]) with entry genes of different viral species in “Th2 high vs. healthy control” gene expression analysis (Figure 2c). Our analysis suggested that genes corresponding to different viral species may co-express with markers of different epithelial subsets. In particular, ACE2 and ANPEP were co-expressed with markers of club, tuft, ciliated cells, and submucosal glands, all downregulated during Th2 high (but not Th2 low) inflammation (Figure 2c). TMPRSS2 expression very closely correlated with mucous ciliated cells, while TMPRSS4, DPP4, and influenza-associated genes correlated with goblet and basal activated cells, all increasing in Th2 high inflammation (Figure 2c). Although we acknowledge that this is only an exploratory correlation analysis, which requires further rigorous validation at the single cell analysis level, it does show the importance of relating viral entry gene expression to epithelial heterogeneity. Single cell studies also support such localization. This explains why there is a complete discrepancy in the expression of ACE2 and TMPRSS2/TMPRSS4 in Th2 high inflammation since these genes differentially follow epithelial subset divergence during inflammation. It is also very important to acknowledge that while ACE2 is specific to certain human coronavirus species (SARS-CoV-1, SARS-CoV-2, and HCoV-NL63), TMPRSS2 and TMPRSS4 are certainly not. These proteases are important for the cleavage of the spike protein of multiple pathogenic viral species, including MERS-CoV [23] and most human influenza strains [24]. 

Of note, the GEO databases used in this study did not identify the ethnicity of participants. Because there is a growing body of evidence supporting the notion that COVID-19 impacts people differentially based on ethnic background [25]. Because various asthma endotypes are also known to associate with ethnic background [26], future investigations into this area should examine the role of ethnicity along with asthma when examining susceptibility to SARS-CoV-2 and other viral diseases.

In summary, changes in epithelial specialization during Th2 high inflammation, as well as sex, location, and disease endotype, may underlie differential susceptibility to SARS-CoV-2 and other human coronavirus species. This warrants further investigation at the single cell level in asthma and other chronic inflammatory conditions.

## Figures and Tables

**Figure 1 viruses-13-01081-f001:**
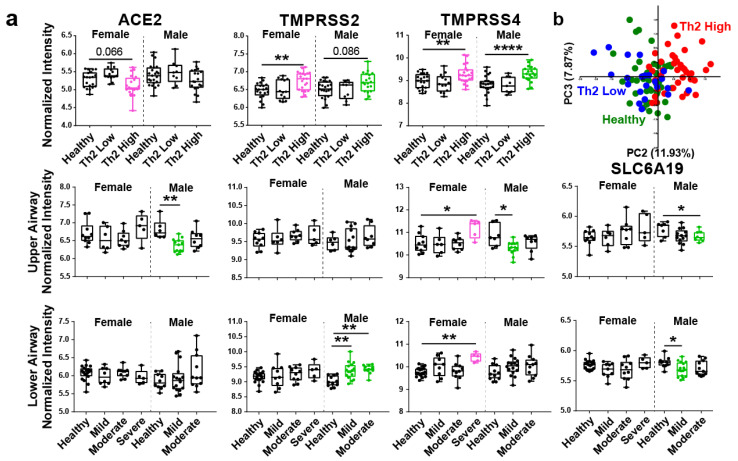
SARS-CoV-2 genes are dysregulated specifically in Th2 high asthma and are different by sex and airway location. (**a**) Box and whisker plots showing normalized intensity values of COVID-19 related genes in healthy and asthmatic samples broken down by sex. Top row compares healthy (*n* = 20 Female, 23 Male), Th2 low (*n* = 13 F, 9 M), and Th2 high (*n* = 2 F, 19 M) asthma. Middle row compares healthy (n = 10 F, 7 M), mild (n = 6 F, 13 M), moderate (n = 8 F, 8 M), and severe (*n* = 5 F) asthma in the upper airway. Bottom row compares healthy (n = 19 F, 11 M), mild (*n* = 9 F, 16 M), moderate (*n* = 10 F, 11 M), and severe (*n* = 5 F) asthma in the lower airway. (**b**) PCA plot summarizes variation in virus entry genes and associated viral immunity genes (ANPEP, DPP4, CTSL, CTSB, ST3GAL4, ST6GAL1, IRAK3, IDO1, NOS2, OAS1, MX1, TNFSF10). Statistical significance was calculated using ANOVA with Tukey post-hoc analysis. In box and whisker plots, all data points are shown and the box represents the second and third quartiles and median. * *p* < 0.5, ** *p* < 0.01, **** *p* < 0.0001.

**Figure 2 viruses-13-01081-f002:**
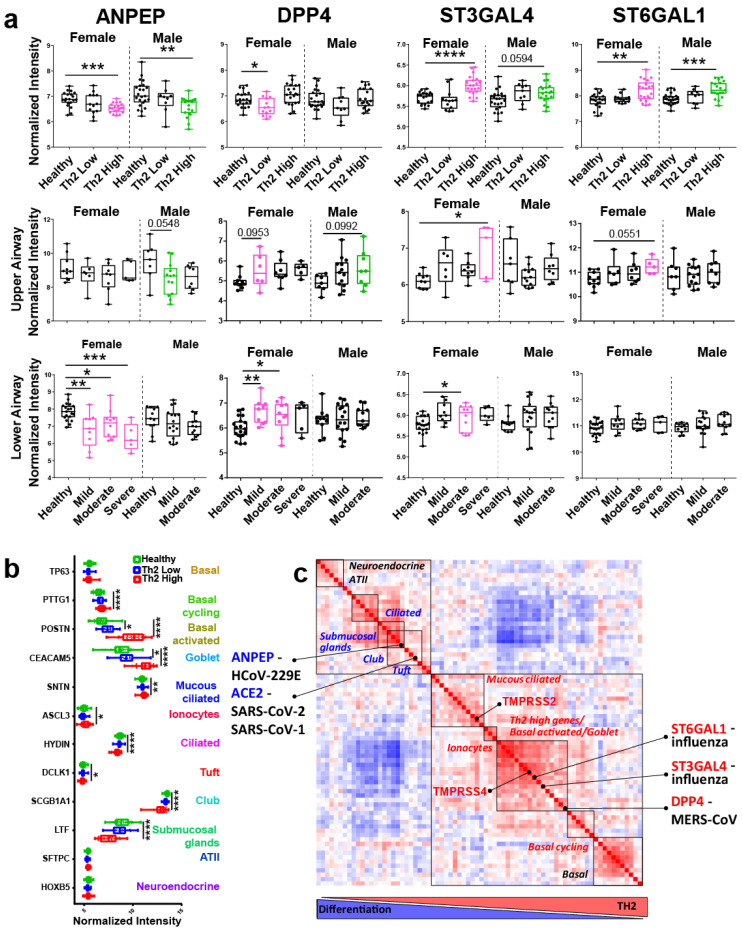
A comparative analysis of human coronavirus and influenza entry genes in Th2 high inflammation. (**a**) Box and whisker plots showing normalized gene expression intensity values of several viral species in healthy and asthmatic samples broken down by sex. Top row compares healthy (*n* = 20 Female, 23 Male), Th2 low (*n* = 13 F, 9 M), and Th2 high (*n* = 2 F, 19 M) asthma. Middle row compares healthy (*n* = 10 F, 7 M), mild (*n* = 6 F, 13 M), moderate (*n* = 8 F, 8 M), and severe (*n* = 5 F) asthma in the upper airway. Bottom row compares healthy (*n* = 19 F, 11 M), mild (*n* = 9 F, 16 M), moderate (*n* = 10 F, 11 M), and severe (*n* = 5 F) asthma in the lower airway. (**b**) Box and whisker plots showing normalized intensity values of key epithelial subset marker genes in healthy, Th2 low, and Th2 high asthma patients. (**c**)**,** similarity matrix demonstrating unsupervised hierarchical clustering of SARS-CoV-1, SARS-CoV-2, HCoV-229E, MERS and influenza entry genes together with top epithelial subset genes from a study by Vieira Braga et al. [8] Higher levels of Th2 inflammation are associated with lower levels of epithelial differentiation (diagram below matrix). Significance in (**a**)**.** was calculated using ANOVA with Tukey post-hoc analysis. All data points are shown in the box and whisker plots, and the box represents the second and third quartiles and median. * *p* < 0.5, ** *p* < 0.01, *** *p* < 0.001, **** *p* < 0.0001.

## Data Availability

All raw data used in this study can be accessed in NCBI GEO under the accession numbers GSE67472 and GSE41861. https://www.ncbi.nlm.nih.gov/geo/ (accessed on 3 June 2021).

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
