# Peer review of "Comparative Study of SARS-CoV-2, SARS-CoV-1, MERS-CoV, HCoV-229E and Influenza Host Gene Expression in Asthma: Importance of Sex, Disease Severity, and Epithelial Heterogeneity"

_viruses, 2021, doi:10.3390/v13061081_

Round 1

Reviewer 1 Report

No additional comments to the revised version.

Author Response

Thank you for the positive review of our revised manuscript.

Reviewer 2 Report

The authors present an in silico study using previously published transcriptomic data from patients with Th2 low versus Th2 high asthma, as well as mild, moderate, and severe asthmatics, to characterize the changes in expression of human coronavirus and influenza viral entry genes relative to sex, airway location, and disease endotype. The results are of interest to researchers and clinicians in trying to understand the mechanisms and/or factors that explain the lack of association of SARS-CoV-2 infection with asthma. However, enthusiasm for the manuscript was low due the lack of description of how their work compares with other published studies related to how asthma status impacts gene expression of SARS-CoV-2 entry genes.

Major criticisms:

  1. In the past year there have been multiple studies investigating the expression of SARS-CoV-2 entry receptors in the context of Asthma which are not cited in this manuscript. One of these studies even uses the GEO dataset GSE67472 listed in this manuscript. Examples manuscripts include:

Wark et al., Respirology. 2021 May;26(5):442-451. doi: 10.1111/resp.14003. Epub 2021 Jan 17. (https://pubmed.ncbi.nlm.nih.gov/33455043/)

Kasela et al., Genome Med. 2021 Apr 21;13(1):66. doi: 10.1186/s13073-021-00866-2. (https://pubmed.ncbi.nlm.nih.gov/33883027/)

Kermani et al., Respir Res. 2021 Jan 7;22(1):10. doi: 10.1186/s12931-020-01605-8. (https://pubmed.ncbi.nlm.nih.gov/33413387/)

Sharif-Askari et al., Mol Ther Methods Clin Dev. 2020 May 22;18:1-6. doi: 10.1016/j.omtm.2020.05.013. (https://pubmed.ncbi.nlm.nih.gov/32537478/)

The authors need to include a description of these studies in their manuscript. Furthermore, they need to emphasize the novelty/originality of their work in the context of these and other studies that investigate the expression of SARS-CoV-2 entry receptors in the context of Asthma.

Text issues

  1. Throughout the manuscript the authors inter-changeably use the terms “SARS-CoV-2” and “COVID19” when discussing the virus. For example, on line 206 the authors describe the “cell subsets expressing the COVID-19 receptor”. This should be “SARS-CoV-2 receptor”. There are numerous examples of this throughout the text which need to be corrected.
  2. The text from lines 185-196 describing the expression of ACE2 in the airway epithelium from mice, humans and macaques is confusing. I would suggest rewriting this and just focusing the section on the human airway/lung which is the focus on this paper.

Author Response

POINT-BY-POINT RESPONSE TO REVIEW COMMENTS

The authors present an in silico study using previously published transcriptomic data from patients with Th2 low versus Th2 high asthma, as well as mild, moderate, and severe asthmatics, to characterize the changes in expression of human coronavirus and influenza viral entry genes relative to sex, airway location, and disease endotype. The results are of interest to researchers and clinicians in trying to understand the mechanisms and/or factors that explain the lack of association of SARS-CoV-2 infection with asthma. However, enthusiasm for the manuscript was low due the lack of description of how their work compares with other published studies related to how asthma status impacts gene expression of SARS-CoV-2 entry genes.

Response: We would like to thank the reviewer for constructive criticism of our manuscript. We carefully addressed all comments and included the suggested citations. We also updated our title that we think now better reflects the scope of work done and emphasizes the novelty of this study. Hopefully, this will now make our manuscript more suitable for publication in the journal. All changes are highlighted in yellow in the marked version of the manuscript.

Major criticisms:

In the past year there have been multiple studies investigating the expression of SARS-CoV-2 entry receptors in the context of Asthma which are not cited in this manuscript. One of these studies even uses the GEO dataset GSE67472 listed in this manuscript. Examples manuscripts include:

Wark et al., Respirology. 2021 May;26(5):442-451. doi: 10.1111/resp.14003. Epub 2021 Jan 17. (https://pubmed.ncbi.nlm.nih.gov/33455043/)

Kasela et al., Genome Med. 2021 Apr 21;13(1):66. doi: 10.1186/s13073-021-00866-2. (https://pubmed.ncbi.nlm.nih.gov/33883027/)

Kermani et al., Respir Res. 2021 Jan 7;22(1):10. doi: 10.1186/s12931-020-01605-8. (https://pubmed.ncbi.nlm.nih.gov/33413387/)

Sharif-Askari et al., Mol Ther Methods Clin Dev. 2020 May 22;18:1-6. doi: 10.1016/j.omtm.2020.05.013. (https://pubmed.ncbi.nlm.nih.gov/32537478/)

The authors need to include a description of these studies in their manuscript. Furthermore, they need to emphasize the novelty/originality of their work in the context of these and other studies that investigate the expression of SARS-CoV-2 entry receptors in the context of Asthma.

Response: Thank you for suggesting to include these citations, we referenced all these papers and discussed our findings in the context of these studies. What stands our study apart is a special emphasis on sex differences and implication of epithelial heterogeneity in disease as potential factors in differential susceptibility to viral infection. Moreover, it is a comparative study of entry genes of multiple viral species in asthma. We made this clearer in Introduction of the manuscript.

Text issues

Throughout the manuscript the authors inter-changeably use the terms “SARS-CoV-2” and “COVID19” when discussing the virus. For example, on line 206 the authors describe the “cell subsets expressing the COVID-19 receptor”. This should be “SARS-CoV-2 receptor”. There are numerous examples of this throughout the text which need to be corrected.

Response: We agree and thank you for pointing this out. We corrected this in all instances where it was appropriate.

The text from lines 185-196 describing the expression of ACE2 in the airway epithelium from mice, humans and macaques is confusing. I would suggest rewriting this and just focusing the section on the human airway/lung which is the focus on this paper.

Response: We rewrote the text as it was suggested by the reviewer.

This manuscript is a resubmission of an earlier submission. The following is a list of the peer review reports and author responses from that submission.

Round 1

Reviewer 1 Report

Well written manuscript addressing an important topic in the present pandemic concerning the question whether different types of asthma propose a risk of COVID-19

The methods are suitable for investigation of this research objective and relevant analyses are used.

The results are discussed appropriately.

It should be stated which type of influenza virus strains are investigated, both in text and figures.

Minor typos should be corrected.

Reviewer 2 Report

Asthma is yet to be identified and described as a risk factor for SARS-CoV-2 infection and COVID-19 disease. Authors hypothesized that it may be related to downregulation of virus entry receptors on host airway cells in asthmatics. They mined the publicly available gene expression database from asthmatic and healthy individuals and looked for genes that are related to airway cell differentiation and virus entry.

They found that some host genes, e.g., ACE2 used by SARS virus as receptor, were downregulated in females with high Th2 asthma, and this was corelated with loss of host airway differentiation genes.

In my opinion, this study is only relating to the function of asthma not virus infection and speculates that changes in the gene expression due to Th1 and Th2 asthma may be the cause of asthma not being identified as COVID-19 risk factor. The database used was not from both asthmatic and infected individuals, because the expression of many host genes will change when those individuals are exposed to the viruses mentioned.

The level of Th1/Th2/related gene expression was not provided in the figures as control.

Figure legends do not mention number of individuals the data represent.

No metrics have been mentioned in the text that, e.g., by what magnitude the expression of ACE2 changed in two datasets; authors left it to the readers to figure out.

Is there any corelation with the age and ethnicity?  

Authors keep mentioning genes like ACE2, TMPRSS 2/4 as viral genes – these are host genes utilised by viruses for infection.  

Minor (line 70), it is not a convention to mention the journal name (facts will not change if a study is published in a certain journal).